# Antimicrobial Resistance and Virulence Characterization of *Listeria monocytogenes* Strains Isolated from Food and Food Processing Environments

**DOI:** 10.3390/pathogens11101099

**Published:** 2022-09-25

**Authors:** Patryk Wiśniewski, Arkadiusz Józef Zakrzewski, Anna Zadernowska, Wioleta Chajęcka-Wierzchowska

**Affiliations:** Department of Industrial and Food Microbiology, University of Warmia and Mazury, Plac Cieszyński 1, 10-726 Olsztyn, Poland

**Keywords:** *Listeria monocytogenes*, virulence factors, virulence genes, antimicrobial resistance, resistance genes, food, food processing environments

## Abstract

*Listeria monocytogenes* is a particularly foodborne pathogen associated with listeriosis, which can be disseminated in food and food processing environments. This study aimed to determine the serotypes and characteristics of virulence factors and antibiotic resistance among 40 *L. monocytogenes* strains isolated from food (*n* = 27) purchased in Olsztyn (Warmia and Mazury region, Poland) and food processing environments in Poland (*n* = 13). Isolates were assigned to serotypes 1/2a, 1/2c, 3a, and 3c using polymerase chain reaction (PCR). The results showed that serotype 1/2a (66.7%) was the most prevalent among strains from food, and serotype 1/2c (53.8%) among strains from the food processing environments. Five different virulence factors (*hlyA, prfA, inlB, luxS, sigB*) were detected in all isolates from the food processing environments using PCR. The *hlyA* (100.0%), *prfA* (100.0%), and *inlB* (96.3%) were the most prevalent in food strains. Seven (25.9%) of the strains of food and ten (76.9%) strains from the food processing environments showed the ability to form biofilm. The tested isolates were subjected to antibiotic susceptibility testing against 12 antibiotics used in the mitigation of listeriosis, using the disk diffusion method. The most frequent were intermediate resistance and resistance to clindamycin. Twelve (92.3%) strains from the food processing environments, and twenty-three (85.2%) from food were non-susceptible to clindamycin. Generally, antibacterial resistance determinants (*Lde*, *aadB*, *aac(3)-IIa(aacC2)^a^*, *penA*, *mefA*, *lnuA*, *lnuB*, *sulI, sulII*) were detected in sixteen (59.0%) strains from food and four (30.8%) from the food processing environments, by PCR. The most frequent were the *mefA-lnuA* (*n* = 7; 20.0%) and *lnuA* (*n* = 6; 17.1%) genotypes. From this research, we can conclude that virulent and antimicrobial-resistant strains of *L. monocytogenes* are present in food and the food processing environment in Poland, which may pose a potential health risk to consumers. Monitoring for the control of virulent and antimicrobial-resistant *L. monocytogenes* strains in the food system can contribute to effective planning and prevention of their spread.

## 1. Introduction

*Listeria monocytogenes* is a foodborne pathogen isolated from food and food processing environments due to its ability to proliferate over a vast range of adverse environmental conditions encompassing low temperature, low pH, high pressure, and high salt concentration [1]. Listeriosis is characterized by high morbidity and mortality, especially in people from risk groups (including the elderly, pregnant women, newborns, and people with weakened immune systems) [2]. In 2019, listeriosis had the highest rate of hospitalised cases and the highest number of deaths of all zoonoses under EU surveillance [3,4].

The species *L. monocytogenes* is classified into four main evolutionary lineages, and thirteen different serotypes of which three serotypes (1/2a, 1/2b and 4b) are responsible for 95% of human listeriosis cases [5]. The ability of this organism to invade host cells depends on many virulence factors, including the presence of genes that are usually located in clusters located throughout the chromosome (including several listerial pathogenicity islands (LIPI-1to LIPI-4) [5,6]. Several virulence genes are located on LIPI-1 (especially *prfA* and *hly*), which are used to assess the virulence potential of isolates [7]. The majority of *L. monocytogenes* isolates belong to Lineages I and II, including serotypes 1/2a, 1/2b, 1/2c, and 4b, which are mostly present in food, natural, and agricultural environments, but have also been isolated from cases of listeriosis in animals and humans. However outbreaks of listeriosis in humans are mostly associated with Lineage I isolates [8].

Due to its virulence and ease of spread in the environment, *L. monocytogenes* remains a serious threat to food safety [1]. Undoubtedly, environmental stress during food production has a significant impact on pathogenicity, gene expression, and changes in antimicrobial resistance. Exposure to sub-lethal concentrations of antimicrobials results in adaptation to higher concentrations of antimicrobials, which may consequently result in cross-resistance to antibiotics [9]. *L. monocytogenes* strains are naturally resistant to cefotaxime, cefepime, fosfomycin, oxacillin, and lincosamides [10,11]. According to other studies, *L. monocytogenes* is at most susceptible to a wide range of antibiotics, which have bactericidal effects against gram-positive bacteria [12]. Unfortunately, recently resistant strains have been observed more frequently among food and environment isolates [9]. Due to adaptive mechanisms, including the exchange of antimicrobial resistance determinants with other bacterial species through horizontal gene transfer, biofilm-forming ability, and efflux pumps, *L. monocytogenes* strains can acquire resistance to the antimicrobials used, resulting in their ineffectiveness [9].

As antimicrobial resistance among microorganisms is growing at an exponential rate and resistance to commonly used antibiotics is spreading among pathogens, it is necessary to better understand the antibiotic resistance and virulence among the strains with various phenotypic and genetic backgrounds, and isolated from various locations in different countries and globally, for further effective mitigation. Therefore, the study aimed to determine the serotypes, the frequency of virulence factors such as the ability to biofilm formation, slime production, and virulence-related genes among *L. monocytogenes* strains isolated from the food and food processing environments in Poland. Moreover, the antimicrobial resistance profile and genes associated with antimicrobial resistance were also examined.

## 2. Materials and Methods

### 2.1. Strains

Isolates were obtained from the microbiological collection of the Department of Industrial and Food Microbiology of the University of Warmia and Mazury in Olsztyn, previously isolated during the pilot study (Appendix A). All isolates were stored in Microbank^TM^ (Biomaxima, Lublin, Poland) at −80 °C. All stored strains were resuscitated by streaking the beads on TSA (Tryptone Soya Agar; Merck, Darmstadt, Germany) and incubation at 37 °C for 24 h. Forty *L. monocytogenes* strains (twenty-seven food strains obtained from foods products purchased in Olsztyn (Warmia and Mazury region Poland) from 2020–2021 and thirteen from the food processing environments obtained from Polish food processing companies from 2020–2021) were used in the current study. Each of the isolates was previously identified phenotypically on ALOA agar (Agar Listeria Ottaviani and Agosti, Merck, Germany). Additionally, all of the strains were confirmed using two methods—the MALDI-TOF MS method (Matrix-Assisted Laser Desorption/Ionisation Time-of-Flight Mass Spectrometry) according to the previously described paper [13], and the genotypic method (polymerase chain reaction (PCR))–described previously by Ryu et al. [14] (Appendix A). PCRs were performed using 2 µL of template DNA, 0.2 µL of *prs* (100 pmol), 0.56 µL of *lmo1030* (100 pmol), 2.5 µL of 10 × DreamTaq PCR Buffer including MgCl_2_ (Thermo Fisher Scientific, Waltham, MA, USA), 0.25 µL of dNTP and 1 U of Taq DNA polymerase (Thermo Fisher Scientific, Waltham, MA, USA) in a total reaction volume of 25 µL. The primer sequences, PCR cycling condition and product sizes are shown in Appendix A.

### 2.2. Serotyping

Total DNA was extracted according to the instruction manuals of the Genomic Mini DNA extraction kit (A&A Biotechnology, Gdynia, Poland). The multiplex PCR technique previously described by Nho et al. (2015) [15] was used for the classification of *L. monocytogenes* strains to different serotypes from Lineage I—most prevalent among food and food-related sources. PCRs were performed using 2 µL of template DNA, 0.4 µL of *flaA* (100 pmol), 0.2 µL of *LMOSLCC2372_0308* (100 pmol) and *LMLG_0742* (100 pmol) and 5 µL of 2 × DreamTaq PCR Master Mix (Thermo Fisher Scientific, Waltham, MA, USA) in a total reaction volume of 25 µL. The primer sequences, PCR cycling condition and product sizes are shown in Appendix A.

### 2.3. In Vitro Biofilm Production Analysis

#### 2.3.1. Detection of the Ability to Slime Production with Congo Red Agar (CRA) Method

The ability to produce slime was determined using the Congo Red Agar method [16]. Plates were streaked with fresh, 24 h culture on CRA and incubated at 37 °C for 24 h. After incubation, the plates were stored at room temperature (23 °C) for 48 h. The ability to produce slime was interpreted according to the colony phenotype. Black colonies were recorded as positive results for slime production, and dark red and red colonies were recorded as negative results.

#### 2.3.2. Biofilm Production Assay by the Microtiter Plate (MTP) Method

The ability to produce a biofilm was tested on 96-well, flat-bottomed, sterile polystyrene plates (Promed^®^) based on the techniques previously proposed by Stepanović et al. [17] with minor changes described previously [18]. The strength of biofilm formation was determined by measuring the absorbance at 570 nm using spectrophotometric microplate reader Varioscan LUX (Thermo Scientific, San José, CA, USA). Optical densities (ODs) for each test strain were determined from the arithmetic mean of 3 replicates by taking measurements at 20 locations in each well. The value obtained was compared with the OD cut-off value (ODc), which was defined as three standard deviations above the mean OD of the negative control. Negative control wells contained BHI broth (Merck, Darmstadt, Germany) only. A scale proposed by Stepanović et al., 2007 [17] was used to determine the strain’s ability to form a biofilm—no biofilm production (OD ≤ ODc), weak biofilm production (ODc < OD ≤ 2xODc), moderate biofilm production (2xODc < OD ≤ 4xODc) and strong biofilm production (4xODc < OD).

### 2.4. The Presence of the Virulence-Associated Genes

All isolates were tested for the presence of five virulence-related genes. The presence of the virulence-associated genes was detected by the PCR technique, previously described by Zakrzewski et al. (2020) [19]. The PCRs were performed using 2 µL of template DNA, 2 µL of each primer (100 pmol), and 12.5 µL of 2 × DreamTaq PCR Master Mix (Thermo Fisher Scientific, Waltham, MA, USA) in a total reaction volume of 25 µL. The primer sequences, PCR cycling conditions, and product sizes of virulence-associated genes are shown in Appendix A. 

### 2.5. Phenotypic Antibiotic-Resistance Analysis

#### Antibiotic Susceptibility Testing and Minimal Inhibitory Concentration (MIC) Determination

Antimicrobial susceptibility was determined using the Kirby–Bauer disc diffusion method according to the standard procedure described by the Clinical and Laboratory Standards Institute (CLSI) [20] and the European Committee on Antimicrobial Susceptibility Testing (EUCAST) [21]. Twelve antibiotics commonly used in the treatment of clinical infection or agricultural procedures were tested. Twelve tested antibiotics (Oxoid, United Kingdom) were selected among ten classes of antimicrobials: aminoglycosides: gentamicin (10 µg); beta-lactams: ampicillin (AMP—10 µg), penicillin G (P—1 U); carbapenems: meropenem (MEM—10 µg); fluoroquinolone: ciprofloxacin (CIP—5 µg); lincosamides: clindamycin (DA—2 µg); macrolides: erythromycin (E—15 µg), vancomycin (VA—30 µg); phenicols: chloramphenicol (C—30 µg); rifampicins: rifampicin (RD—5 µg); sulphonamides: trimethoprim-sulfamethoxazole (SXT—25 µg); tetracycline: tetracycline (TE—30 µg).

McFarland (0.5) standard concentration suspensions in sterile saline (0.9%) were prepared from overnight bacterial colonies on TSA (Merck, Germany). A sterile swab was used to inoculate the suspension evenly on Mueller-Hinton agar (Merck, Darmstadt, Germany), and antibiotics discs were transferred on the agar using a disc dispenser, after which they were incubated for 24 h at 37 °C. After incubation, the inhibition zones were measured and interpreted as “Resistant (R), Intermediate (I), or Susceptible (S)” using standard reference values according to EUCAST for *L. monocytogenes* [21]. Additionally, due to the lack of standards for antibiotics not included in EUCAST for *L. monocytogenes,* the standards for Staphylococci were used.

Strains that were identified as resistant to antibiotics by the Kirby–Bauer diffusion method (Resistant (R), Intermediate (I)) were tested for Minimal Inhibitory Concentration (MIC) using MTS^TM^ (MIC Test Strips) (Liofilchem^®^, Roseto degli Abruzzi, TE, Italy) following the manufacturer’s instructions. This study included five different antibiotics: ciprofloxacin, gentamicin, clindamycin, penicillin G, and SXT (trimethoprim-sulfamethoxazole). 

### 2.6. Detection of Antimicrobial Resistance Genes

The isolates showing resistance and intermediate resistance obtained from phenotypic analyses were selected for antimicrobial resistance genes detection analyses. Isolates showing susceptibility in phenotypic analysis were not selected for further analyses. 

Antimicrobial resistance genes were screened using PCR techniques. Eight antimicrobial resistance genes were tested, including genes encoding resistance to ciprofloxacin (*Lde*), gentamicin (*aadB, aac(3)-IIa(aacC2)^a^*), penicillin G (*penA*), clindamycin (*lnuA, lnuB, mefA*), and trimethropim/sulfametaxazole (*sulI, sulII*). Primer sequences, concentration, PCR cycling condition, and product sizes of antimicrobial resistance genes are shown in Appendix A.

### 2.7. Statistical Analysis

Statistical analysis was performed using STATISTICA 13.3 StatSoft^®^ software (StatSoft Inc., Tulsa, OK, USA) using the Chi-squared test. For the analyses, *p* ≤ 0.05 was considered significant.

## 3. Results

### 3.1. Presence of Serotypes and Virulence Factors of L. monocytogenes Isolates

Isolates were assigned to serotypes 1/2a, 1/2c, 3a, and 3c. Serotype 1/2a (66.7%) was the most prevalent among strains from food samples, and serotype 1/2c (53.8%) among strains from the food processing environments. Three strains isolated from food were not assigned to lineage I strains. Five virulence determinants were also characterized in the study. The genes belonging to LIPI-1, *prfA,* and *hlyA* were detected in all isolates. The *inlB* gene was the most prevalent in food isolates. In isolates from the food processing environments, *inlB, luxS, and sigB* genes were detected in all isolates. The results showed that only two of the food isolates and one strain from the food processing environments showed a strong ability to form biofilm in the microtiter plate (MTP) method. Four strains from both the food and food processing environments showed a moderate ability for biofilm production. One isolate from food and five isolates from the food processing environments showed a weak ability to produce biofilm. The statistical analysis showed that the isolation source of *L. monocytogenes* has a significant effect on the isolated serotype (*p* = 0.0009) and ability to produce biofilm (*p* = 0.0022). No statistically significant differences were found between the source of isolation and the ability to produce slime (*p* = 0.1435) and resistance to antimicrobial agents (*p* = 0.6972).

Evaluation of the ability of slime production using the CRA method showed that only four strains from food are slime producers. None of the food processing environment strains showed the ability to produce slime. The virulence characteristics of the tested strains are presented in Table 1. The results for individual isolates and various tests in this study were provided together in the Appendix A.

### 3.2. Antimicrobial Susceptibility Testing and Minimal Inhibitory Concentration (MIC) Determination

In total, 37 isolates (92.5%) were resistant or intermediately resistant to one or more antibiotics. All strains were sensitive to the following antibiotics: ampicillin, chloramphenicol, erythromycin, rifampicin, tetracycline, and vancomycin. A high frequency of resistance (R) and intermediate resistance (I) to clindamycin among the strains were observed. Some of the strains from food also showed resistance to meropenem and trimethoprim/sulfamethoxazole. In addition, strains from food processing environments showed resistance to penicillin G and trimethoprim/sulfamethoxazole. Additionally, strains from food also revealed intermediate resistance to ciprofloxacin and gentamicin. The analyses were performed against 12 antibiotics, and for each analysis, *L. monocytogenes* strain is summarised in Table 2.

The isolates showed seven different antibiotic resistance patterns (Table 3). The most observed pattern was resistance and intermediate resistance to clindamycin. A total of 20 isolates (74.0%) from foods and ten isolates (77.0%) from the food processing environments were resistant or intermediately resistant to only one antimicrobial agent. Only one isolate (3.7%) from food showed a multiresistance pattern.

### 3.3. Antimicrobial Resistance Gene Profiling

Isolates showing phenotypic antimicrobial resistance (R or I) were tested for the presence of the gene encoding this resistance. Ciprofloxacin resistance gene *Lde* and gentamicin resistance gene *aadB* were detected in all strains showing phenotypic resistance. Sulphonamide resistance gene *sulI* was detected in four (80.0%) of five isolates, whereas the *sulII* gene was detected in only two strains (40.0%). Clindamycin resistance genes were detected in 15 (42.9%) of 35 strains. The *lnuA* gene was most prevalent among the strains (*n* = 13, 37.2%) compared with the *mefA* gene (*n* = 9, 25.8%). The *lnuB*, *penA,* and *aac(3)-IIa(aacC2)^a^* genes were not detected in any of the isolates (Table 4). The strains showed three different resistance genotypic patterns for clindamycin (Table 5). Generally, tested resistance determinants were detected in 15 (55.6%) strains from food and four (30.8%) strains from the food processing environments. A total of eight resistant isolates (50.0%) from foods and two resistant isolates (40.0%) from the food processing environments had only one tested resistance gene. 

## 4. Discussion

*Listeria monocytogenes* is the main aetiology of listeriosis, a serious threat to human health [22]. In this study, all four serotypes that belong to Lineage I were observed. The most prevalent serotypes were serotypes 1/2a (66.7%) for food and 1/2c (53.8%) for the food processing environments. The results were consistent with previously published studies [23,24,25,26,27,28]. In food and the production environment, the dominant group is lineage I (1/2a, 1/2c, 3a, 3c) due to its potential ability to survive in many food matrices. Furthermore, serotypes belonging to lineage I are one of the causes of outbreaks of listeriosis in humans.

The pathogenicity of *L. monocytogenes* is affected by various virulence determinants whose presence may significantly contribute to an increase in the strains’ virulence, and thus, cause infections in humans [29]. This study analysed five different factors that encode the strains’ virulence. In each isolate, the presence of the LIPI-1 pathogenicity island was confirmed. The LIPI-1 island comprises six genes (*prfA*, *plcA*, *hly*, *mpl*, *actA,* and *plcB*), of which two, *prfA* and *hly*, are mainly identified as target genes in the assessment of the virulence potential of *L. monocytogenes* isolates in view of their possible transcription during the development of these pathogens in different types of food [7]. The PrfA transcription activator, encoded by *prfA*, is responsible for the transcription of more than 140 genes (including all the genes present in LIPI-1). Listeriolysin O (LLO), a toxin required for the bacterial escape from phagosomes is encoded by *hly*. The absence of LLO results in the strain’s avirulence [30]. In the current study analysed the presence of two LIPI-1 genes, i.e., *prfA* and *hly*, which were identified in all the tested strains. These results are in line with those obtained in studies conducted by Sudagidan et al., 2021 [31] and Iwu and Okoh, 2020 [32], who demonstrated the presence of the *hlyA* gene in all strains (*n* = 32 and *n* = 20, respectively), and in a study by Du et al., 2016 [33], who detected the *prfA* and *hlyA* genes in 100.0% of the tested strains (n = 21). 

In addition to the LIPI-1 island genes, the present study also analysed three genes (*inlB*, *luxS*, *sigB*) that affect the biofilm formation ability. The analysed genes were detected in all the tested strains derived from the food processing environment, while in the strains derived from food, the *inlB* gene was detected in almost all the strains (*n* = 26; 96.3%), while the *luxS* and *sigB* genes were detected in 20 strains (74.1%). The presence of the *inlB* gene (internalin B encoding gene) is linked to the strain’s ability to adhere to abiotic surfaces [34]. This gene was identified in all *L. monocytogenes* strains isolated from food as well as from the environment [6,31,35]. However, in certain studies, the prevalence is lower (*n* = 15; 71.4%) [33]. The *luxS* gene (which encodes the production of the autoinductor-2 (AI-2) molecules) is responsible for the interspecies communication and is also specifically involved in the ability of biofilm formation by *L. monocytogenes* [36]. The last of the analysed genes, i.e., *sigB* (responsible for encoding alternative sigma factor sigma B), is one of the two main transcription factors (except PrfA) affecting biofilm formation ability [37]. In the literature, the *luxS* genes are determined with a similar frequency of 71.4% (*n* = 5), while the *sigB* gene has a frequency of 100.0% (*n* = 7) [19]. The differences in the prevalence of individual genes may be due to mutations as well as to the source of isolation and origin of a particular strain [32]. The biofilm provides an ideal environment for growth and development and is one of the main concerns of the food industry [38]. The ability to form biofilm is a characteristic that facilitates *L. monocytogenes* survival of adverse environmental conditions in both food and the food production environment [39]. The strains derived from food processing environments were more frequently able to form a biofilm stronger than those generated by the strains isolated from food. The high prevalence of *L. monocytogenes* strains carrying the individual virulence genes obtained in the current study may indicate the high potential pathogenicity of the analysed strains and show that the isolates derived from food and production facilities may pose a real hazard to the public health [40]. The isolates derived from food processing environments are characterized by a greater number of various virulence factors than the strains derived from food due to their exposure to many sublethal stress factors such as osmotic stress, high hydrostatic pressure and acid stress prevailing in food production facilities, which improves their survivability and increases pathogenicity [32,41].

The published studies have reported the increasing of *L. monocytogenes* strains’ resistance to the entire range of antibiotics of different classes [27,32,35,42,43,44,45,46,47,48]. The study analysed resistance to 12 antimicrobial agents, and demonstrated that the analysed *L. monocytogenes* strains were susceptible to a considerable proportion of the analysed agents except clindamycin, meropenem, penicillin G, and trimethoprim/sulfamethoxazole. Low resistance to meropenem (*n* = 1; 2.5%) and trimethoprim/sulfamethoxazole (*n* = 5; 12.5%) is in line with the results obtained previously by Şanlıbaba et al., 2018 [49] (resistance to meropenem in one strain (5.9%) and to trimethoprim/sulfamethoxazole in three isolates (17.7%)), and by Aksoy et al., 2018 [50] (resistance to meropenem in one strain (6.7%), and to trimethoprim/sulfamethoxazole in four (26.7%)). The resistance to trimethoprim/sulfamethoxazole is related to the presence of the genes *sulI* and *sulII* [51]. The *sulI* gene was detected in four (80.0%) out of five strains that exhibited resistance to trimethoprim/sulfamethoxazole, while the *sulII* gene was detected in two (40.0%) of the strains. Both genes were determined in one of the isolates derived from the food processing environment. As regards the resistance to penicillin G, resistance was noted in only one (2.5%) of the tested isolates. The results in the current study are in line with the observations of Noll et al., 2018 [52], who noted three (1.2%) strains resistant to penicillin G, and of Li et al., 2016 [53], who determined one (1.3%) resistant strain. In the current study, the phenotypic profile of *L. monocytogenes* resistance towards penicillin G was not confirmed through the detection of the *penA* gene. A study conducted by Babarandage et al., 2022 [54] also noted no presence of the *penA* gene in the isolates exhibiting phenotypic resistance to penicillin G.

Phenotypic antibiotic resistance testing indicated a particularly high level of resistance (*n* = 12; 30.0%) and intermediate resistance (*n* = 23; 57.5%) one of the antibiotics tested, i.e., clindamycin. Clindamycin is widely used in hospital and veterinary treatment in the event of infections caused by gram-positive bacteria [55]. Its mechanism of action is similar to that of erythromycin, i.e., it is responsible for the inhibition of protein synthesis due to the binding with the 50S subunit of bacterial ribosomes [49,56]. There are several studies indicating a high level of clindamycin resistance among the *L. monocytogenes* isolates derived from food and production facilities [45,53,54,55,56,57,58,59,60,61]. According to studies conducted by Şanlıbaba et al., 2018 [49] and Escolar et al., 2017 [56], the strains’ resistance to clindamycin may be associated with the action of the clindamycin structure-modifying enzyme that contributes to the inactivation of the antibiotic action. The strains’ resistance to clindamycin is often combined with sensitivity to erythromycin. In the current study, all strains exhibited sensitivity to this antibiotic of the macrolide class. This atypical combination, referred to as the L^R^/M^S^ phenotype (lincosamide resistance and macrolide susceptibility phenotype), has been increasingly noted and may be associated with *lnuA* and *lnuB* genes [56,62]. In the current studies, the *lnuA* gene was detected in 13 strains (37.2%), while the presence of the *lnuB* gene was not found in any of them. In a study by Swetha et al., 2021 [27], the *lnuA* gene was identified in one (10.0%) of the strains isolated from food, while the *lnuB* gene was not detected in any of the isolates. These results are also in line with those obtained in a study by Escolar et el., 2017 [56], who analyzed seven *L. monocytogenes* strains isolated from ready-to-eat food. Each strain exhibited resistance to clindamycin while lacking the *lnuA* and *lnuB* genes. The current study also analyzed the presence of the macrolide resistance gene *mefA*. In contrast to the studies by Escolar et al., 2017 [56] and Granier et al., 2011 [63] (no isolates carrying the *mefA* gene), this gene was detected in nine (22.5%) isolates. Interestingly, seven of them also had the *lnuA* gene, and these strains exhibited intermediate resistance towards clindamycin. The strains isolated from food were characterized by a higher prevalence of the clindamycin resistance-encoding genes (*n* = 12; 56.0%) than the strains from the production environment (*n* = 2; 16.6%). It is presumed that the resistance in the strains having no clindamycin resistance-encoding genes may be associated specifically with the action of the clindamycin structure-modifying enzyme, whose activity may be induced by the effects of the sub-lethal environmental stress prevailing during food production. Further research is necessary to confirm this hypothesis. 

## 5. Conclusions

The study provided data on the phenotypic and genotypic characteristics of serotypes, virulence, and antimicrobial resistance in *L. monocytogenes* isolates isolated from food and food production environments in Poland. This study indicated that the *L. monocytogenes* strains isolated from food purchased in Poland mainly belong to the 1/2a serotype, defined as one of the three serotypes with the highest pathogenic potential, which confirms the results of research on the occurrence of *L. monocytogenes* strains belonging to Lineage I in food and industry. Since the strains belonging to the 1/2a serotype may be more virulent than others, it is necessary to conduct additional research in order to assess their virulence. The isolates isolated from the food processing environment obtained from Polish plants were characterized by a higher frequency of biofilm formation, with the formed biofilms being stronger than those generated by the strains from food. The tested strains, in accordance with results on other strains isolated from food most frequently exhibited resistance to clindamycin. These results indicate resistance of the strains to antibiotics (penicillin G, gentamicin, ciprofloxacin, meropenem, trimethoprim/sulfamethoxazole) applicable to the clinical treatment of the disease. Likewise, the results obtained expand the knowledge of the pathogen’s prevalence in the study region and point to an important public health problem that threatens the health of consumers. Therefore, the surveillance and monitoring of the virulence and antibiotic resistance of the *L. monocytogenes* strains is becoming important due to the increasing number of antibiotic-resistant strains isolated from food and industry.

## Figures and Tables

**Table 1 pathogens-11-01099-t001:** Occurrence, serotypes, phenotypic, and genotypic determinants of *L. monocytogenes* virulence recovered from food and food processing environments.

Sample Type			Food	Food Processing Environments
**No. of *L. monocytogenes* Strains**			27 (67.5)	13 (32.5)
**Serotypes (%)**		**1/2a**	18 (66.7)	2 (15.4)
		**1/2c**	2 (7.4)	7 (53.8)
		**3a**	0 (0.0)	2 (15.4)
		**3c**	4 (14.8)	2 (15.4)
		**Other**	3 (11.1)	0 (0.0)
**Genetic** **determinants** **of virulence (%)**	**LIPI-1**	** *hlyA* **	27 (100.0)	13 (100.0)
		** *prfA* **	27 (100.0)	13 (100.0)
	**Biofilm**	** *inlB* **	26 (96.3)	13 (100.0)
		** *luxS* **	20 (74.1)	13 (100.0)
		** *sigB* **	20 (74.1)	13 (100.0)
**Biofilm** **formation (%)**		**Weak**	1 (3.7)	5 (38.5)
		**Moderate**	4 (14.8)	4 (30.8)
		**Strong**	2 (7.4)	1 (7.6)
		**Negative**	20 (74.1)	3 (23.1)
**Slime** **production (%)**		**Positive**	4 (14.8)	0 (0.0)
		**Negative**	23 (85.2)	13 (100.0)

**Table 2 pathogens-11-01099-t002:** Antibiotic resistance of *L. monocytogenes* strains isolated from food and food processing environments.

No.	Antibiotic	Resistance	Food		Food Processing Environments		Total
			Test Method Used				
			Disc Diffusion (%)	MIC Range (µg/mL)	Disc Diffusion (%)	MIC Range (µg/mL)	
1.	AMP	S	27 (100.0)	ND	13 (100.0)	ND	40 (100.0)
		I	0	ND	0	ND	0
		R	0		0		0
2.	C	S	27 (100.0)	ND	13 (100.0)	ND	40 (100.0)
		I	0	ND	0	ND	0
		R	0		0		0
3.	CIP	S	25 (92.6)	ND	13 (100.0)	ND	38 (95.0)
		I	2 (7.4)	0.38–0.50	0	ND	2 (5.0)
		R	0		0		0
4.	E	S	27 (100.0)	ND	13 (100.0)	ND	40 (100.0)
		I	0	ND	0	ND	0
		R	0		0		0
5.	CN	S	26 (96.3)	ND	13 (100.0)	ND	39 (97.5)
		I	1 (3.7)	0.19	0	ND	1 (2.5)
		R	0		0		0
6.	DA	S	4 (14.8)	ND	1 (7.7)	ND	5 (12.5)
		I	14 (51.9)	1.0–32.0	9 (69.2)	1.5–4.0	23 (57.5)
		R	9 (33.3)		3 (23.1)		12 (30.0)
7.	MEM	S	26 (96.3)	ND	13 (100.0)	ND	39 (97.5)
		R	1 (3.7)	0.047	0		1 (2.5)
8.	P	S	27 (100.0)	ND	12 (92.3)	ND	39 (97.5)
		R	0		1 (7.7)	1	1 (2.5)
9.	RD	S	27 (100.0)	ND	13 (100.0)	ND	40 (100.0)
		I	0		0		0
		R	0		0		0
10.	SXT	S	25 (92.6)	ND	10 (76.9)	ND	35 (87.5)
		R	2 (7.4)	0.064	3 (23.1)	0.064–0.125	5 (12.5)
11.	TE	S	27 (100.0)	ND	13 (100.0)	ND	40 (100.0)
		I	0		0		0
		R	0		0		0
12.	VA	S	27 (100.0)	ND	13 (100.0)	ND	40 (100.0)
		I	0		0		0
		R	0		0		0

MIC—minimum inhibitory concentration by using MTSTM (MIC Test Strips); ND—Not detected. S—Susceptible, I—Intermediate, R—Resistance; AMP—Ampicillin, C—Chloramphenicol, CIP—Ciprofloxacin, E—Erythromycin, CN—Gentamicin, DA—Clindamycin, MEM—Meropenem, P—Penicillin G, RD—Rifampicin, SXT—Trimethoprim/Sulfamethoxazole, TE—Tetracycline, VA—Vancomycin.

**Table 3 pathogens-11-01099-t003:** Antimicrobial resistance profiles of *L. monocytogenes* strains by serotype.

		Food					Food Processing Environments				
		Number (%) of Resistant Strains (n = 27)					Number (%) of Resistant Strains (n = 13)				
		1/2a	1/2c	3c	Other	Total	1/2a	1/2c	3a	3c	Total
1.	**DA (R/I)**	13 (48.1)	1 (3.7)	3 (11.1)	3 (11.1)	20 (74.0)	2 (15.4)	4 (30.8)	2 (15.4)	2 (15.4)	10 (77.0)
3.	**CN (I), SXT (R)**	1 (3.7)	-	-	-	1 (3.7)	-	-	-	-	-
4.	**DA (I), CIP (I)**	1 (3.7)	-	1 (3.7)	-	2 (7.4)	-	-	-	-	-
5.	**DA (I), SXT (R)**	-	-	-	-	-	-	2 (15.4)	-	-	2 (15.4)
6.	**P (R), SXT (R)**	-	-	-	-	-	-	1 (7.7)	-	-	1 (7.7)
7.	**DA (R), MEM (R), SXT (R)**	1 (3.7)	-	-	-	1 (3.7)	-	-	-	-	-
**Not resistance for all**		2 (7.4)	1 (3.7)	-	-	3 (11.1)	-	-	-	-	-

S—Susceptible, I—Intermediate, R—Resistance; CIP—Ciprofloxacin, CN—Gentamicin, DA—Clindamycin, MEM—Meropenem, P—Penicillin G, SXT—Trimethoprim/Sulfamethoxazole.

**Table 4 pathogens-11-01099-t004:** The occurrence of resistance genes of *L. monocytogenes* strains isolated from the food and food processing environments.

Antibiotic (n)	Antibiotic Resistance Genes	Isolates from Food	Isolates from the Food Processing Environments	Total Positive
CIP (2)	*Lde*	2 (100.0)	-	2 (100.0)
CN (1)	*aadB* *aac(3)-IIa(aacC2)^a^*	1 (100.0) 0 (0.0)	- 0 (0.0)	1 (100.0) 0 (0.0)
P (1)	*penA*	0 (0.0)	0 (0.0)	0 (0.0)
DA (35)	*mefA* *lnuA* *lnuB*	8 (22.9) 12 (34.3) 0 (0.0)	1 (2.9) 1 (2.9) 0 (0.0)	9 (25.8) 13 (37.2) 0 (0.0)
SXT (5)	*sulI* *sulII*	1 (20.0) 1 (20.0)	3 (60.0) 1 (20.0)	4 (80.0) 2 (40.0)

CIP—Ciprofloxacin, CN—Gentamicin, P—Penicillin G, DA—Clindamycin, SXT—Trimethoprim/Sulfamethoxazole. “a” is a continuation of the gene name

**Table 5 pathogens-11-01099-t005:** Genotypic resistance patters between *L. monocytogenes* strains by serotype for clindamycin.

		Food					Food Processing Environments					Total Number (%) of Clindamycin Resistant Strains (n = 35)
		Number (%) of Clindamycin Resistant Strains (n = 23)					Number (%) of Clindamycin Resistant Strains (n = 12)					
		1/2a	1/2c	3c	Other	Total	1/2a	1/2c	3a	3c	Total	
**1.**	*lnuA*	4 (17.0)	-	-	1 (4.3)	5 (21.3)	-	1 (8.3)	-	-	1 (8.3)	6 (17.1)
**2.**	*mefA*	1 (4.3)	-	-	-	1 (4.3)	-	1 (8.3)	-	-	1 (8.3)	2 (5.7)
**3.**	*mefA-lnuA*	3 (12.7)	-	2 (8.4)	2 (8.4)	7 (30.4)	-	-	-	-	-	7 (20.0)

## Data Availability

The data presented in this study are available on reasonable request from the corresponding author.

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
