# Peer review of "Antimicrobial Resistance and Virulence Characterization of *Listeria monocytogenes* Strains Isolated from Food and Food Processing Environments"

_pathogens, 2022, doi:10.3390/pathogens11101099_

Round 1
Reviewer 1 Report
The content of this paper was technically accurate and sound.
The research methodology for the study was appropriate and applied properly.
BUT;
The introduction did not provide the necessary background information. It should be improved.
Reviewer 2 Report
Line 134 – 2.6. Detection of Antimicrobial Resistance Genes. It is necessary to understand the criteria to select the isolates for the PCR analysis. The authors need to explain that.
Why was the resistance analysis performed only between the isolates resistant and intermediate? The authors need to explain. Sometimes sensitive isolates may show resistance genes without expressing them in the phenotypic analysis.
Line 166 – The authors should change the word “Incidence”. I’m not sure if the result supports the use of this term. Maybe the authors could think about changing to ‘Frequency”, for example. By the way, I suggest the authors make some changes to this paragraph regarding the epidemiological definitions.
Information on the production environments in which the strains were isolated is lacking. In how many locations were the samples collected? Is it a single industry? What foods were processed in each?
Reviewer 3 Report
General: Organized, and showed interesting results which are worthwhile for publication. However the numbers (n=40) of the isolates tested were small. Therefore, it is important to define the source of the isolates which may provide or support the published studies in other regions of the world or a particular country, and how the information from this small numbers of the samples can be useful. There are also many grammar errors or issues that need to be corrected or improved. The comments below only address some of them. The following are particular comments.
1. Abstract
1.1.General: Need to define the rational and objective(s) of the study, namely why examined the antibiotic resistance and virulence factors in 40 L. monocytogenes isolates? Many published studies have been carried out on these aspects internationally, why the current study was necessary? Was this to further investigate the isolates from a particular region or country and to provide risk information for further detection and prevention purposes? If so need to provide the name of the region from which the strains were isolated. And to provide the conclusion relevant to the objective(s).
1.2.Line 11. Change the “within the food and food processing environments” to “… in food and food processing environments.”
1.3.Lines 11-13. Change “This study investigated to determine. …” to “This study aimed to determine ….:”, change “the serotypes and characteristics of virulence factors and antibiotic resistance among 40 L. monocytogenes strains isolated from the food (n=27) and food processing environments 13 (n=13).
1.4.Line 15. Delete the word “obtained”.
2. Introduction:
2.1. As mentioned for the abstract, it is not clear why the study is carried out while so many similar studies have already published. Therefore, need to define the rational and objective(s) of the study, namely why examined the antibiotic resistance and virulence factors in 40 L. monocytogenes isolates? Many published studies have been carried out on these aspects internationally, why the current study was necessary? Was this to further investigate the isolates from a particular region or country and to provide risk information for further detection and prevention purposes? If so need to provide the name of the region from which the strains were isolated. And to provide the conclusion relevant to the objective(s). Why a few tests including slime production biofilm formation, detection of antibiotics resistant genes, were used, how these tests together will help to reach the goal.
2.2. Please consider to provide brief review on the Listeria monocytogenes lineages, and serogroups, and the relationship between the lineages and serogroups.
3. Methods.
3.1. Please provide more details of the food
processing\environment, namely for what type of food, which
stage(s) of the processing, from surface of equipment or floor
etc.
3.2. Need to define how the biofilm formation was
determined, namely the controls and cut-off line, and how
weak, moderate and strong former was determined.
3.3. Consider to use statistical analysis, such as correlation
and/or chi-square analysis to see the correlation between
different factors.
4. Results
4.1.Since the numbers of the isolates tested were small, maybe it is reasonable to add additional table to show all the results for each individual strain. This will help to understand the detailed virulent factors/mechanisms based on a single strain.
4.2. Line 149. Change the sentence “According to the obtained results…” to “The results showed that ….”.
5. Discussion
5.1. Please discuss how the virulence factors and antibiotics resistance enhance the pathogenicity of the L. monocytogenes isolates.
5.2. Lines 235-236. Change the phrase “authors’ own study” to “The present study”.
6. Conclusion:
6.1. should provide the significance of this study including scientific or technical contributions.
6.2. Lines 327-329. Reword the sentence “These results are worrying, considering their use in clinical treatment. Therefore, the surveillance and monitoring of the virulence and antibiotic resistance of the L. monocytogenes strains is becoming important due to the increasing number of antibiotic- resistant strains isolated from food and industry”. For example, the meaning of the word “worrying” is not clear in this sentence.
Round 2
Reviewer 1 Report
It can be published in present form.
Reviewer 3 Report
Authors have made good efforts to address the reviewer's comments and suggestions. Following are only suggestions for the minor editorial changes.
1. Line 31, delete the “,” after between “environment, in Poland”.
2.Line 45, please consider to change the phrase “Among the L. monocytogenes, there is a classification into…” to “The species L. monocytogenes is classified into…”
3. Line 47, add “this organism after “The ability to” as “The ability of this organism”.
4. Line 48, change”among which are” to “including…”
5. Lines 53 and 54: change “were also” to “have also been”.
6. Line 61, delete “as”.
7. Line 62, Change “resistance” to “resistant”.
8. Line 68, add reference for the sentence “Unfortunately, recently resistant strains 67 have been observed more frequently among food and environment isolates.”
9. Line 70, change “exchange resistance determinants” to “the exchange of antimicrobial resistance determinants”.
10. Line 72, change the spelling “pomps’ to “pumps”, and add “and” between “ability” and efflux pumps” as “ability and efflux pumps”.
11. Line 73, delete the full stop before the reference.
12. Lines 75-77, modify the sentence “it is undoubtedly necessary to control both phenotypic and genotypic resistance among strains and their virulence”. To “it is necessary to better understand the antibiotic resistance and virulence among the strains with various phenotypic and genetic background and isolated from various locations in different countries and globally for further effective mitigation.”
13. Line 80, add “in Poland” after “environment”.
14. Lines 132, change the phrase “was determined by measuring the absorbance at 570 nm was measured with …” to “was determined by measuring the absorbance at 570 nm using …”
15. Line 136, delete comma after “which was defined,”
16. Line 189, change “2.6. Statistical Analysis” to “2.7….”
17. Line 210, change “statistical” to “statistically”.
18. Lines 215-216, change the sentence “The results of the study were collected in a summary form 215 in Supplementary Table S1.” to “The results for individual isolates and various tests in this study were provided together in the Supplementary Table S1.”
19. Lines 238, 360, and potentially other areas, change “… authors’ own studies”, to “in the current study”.
20. Line 323, change “The ongoing studies report increasing...” to “The published studies have reported the increasing of…”
21. Line 346, change the phrase “to one of them” to “one of the antibiotics tested,”
22. In conclusion, if possible, please briefly summarize what is or are the novel result(s) obtained in this study, such as if the current study provide some new or different results or same or similar to other published studies. This may help others to use the results from this study to better understand the more comprehensive or global picture.
